# PEARL: PROTOTYPE LEARNING VIA RULE LISTS

## ABSTRACT

Deep neural networks have demonstrated promising prediction and classification performance on many healthcare applications. However, the interpretability of those models are often lacking. On the other hand, classical interpretable models such as rule lists or decision trees do not lead to the same level of accuracy as deep neural networks and can often be too complex to interpret (due to the potentially large depth of rule lists). In this work, we present PEARL, Prototype lEArning via Rule Lists, which iteratively uses rule lists to guide a neural network to learn representative data prototypes. The resulting prototype neural network provides accurate prediction, and the prediction can be easily explained by prototype and its guiding rule lists. Thanks to the prediction power of neural networks, the rule lists from prototypes are more concise and hence provide better interpretability. On two real-world electronic healthcare records (EHR) datasets, PEARL consistently outperforms all baselines across both datasets, especially achieving performance improvement over conventional rule learning by up to $28\%$ and over prototype learning by up to $3\%$. Experimental results also show the resulting interpretation of PEARL is simpler than the standard rule learning.

## 1 INTRODUCTION

The rapid growth of sizes and complexities of electronic health records (EHR) data has motivated the use of deep learning models, which demonstrated state-of-the-art performance in many tasks, including diagnostics and disease detection (Lipton et al., 2016; Choi et al., 2016b; Xiao et al., 2018a), medication prediction (Zhang et al., 2017; Le et al., 2018), risk prediction (Futoma et al., 2015; Xiao et al., 2018b), and patient subtyping (Baytas et al., 2017; Che et al., 2017). Although deep learning models can produce accurate predictions and classifications, they are often treated as black-box models that lack interpretability and transparency of their inner working (Lipton, 2016). This is a critical problem as it can limit the adoption of deep learning in medical decision making.

Recently, there have been great efforts of trying to explain black-box deep models, including via attention mechanism (Choi et al., 2016b; Xu et al., 2015), visualization (Samek et al., 2017), and explanation by examples or prototypes (Li et al., 2017). To bring deep models into real clinical practice, clinicians often need to understand why a certain output is produced and how the model generates this output for a given input (Nauck & Kruse, 1999). Rule learning and prototype learning are two promising directions to achieve clinical model interpretability.

Rule learning generates a set of rules from training data, in which its prediction is done at leaf levels via simple models such as majority vote or regression. For example, the results of rule learning are rule lists composed of multiple if-then statements (Angelino et al., 2018). Those rules can be interpretable to domain experts as they are expressed in simple logical forms (Rivest, 1987; Breiman, 2017). However, because of such a simple prediction model, the accuracy of rule-based models is often lower than deep neural networks. Moreover, the interpretability can be undermined as the depth of rules becomes very large and thus incomprehensible for human with tens or hundreds of levels of the rules.

Prototype learning is another interpretable model inspired by case-based reasoning (Kolodner, 1992), where observations are classified based on their proximity to a prototype point in the dataset. Many machine learning models have incorporated prototype concepts (Priebe et al., 2003; Bien & Tibshirani, 2011; Kim et al., 2014), and learn to compute prototypes (as actual data points or synthetic points) that can represent a set of similar points. However prototypes alone may not lead to

interpretable models as we still need an intuitive way to represent and explain what a prototype is, especially given recent deep prototype works (Li et al., 2017).

Both approaches were explored in healthcare applications. For example, rule learning was employed to identify how likely patients were to be readmitted to a hospital after they had been released, each probability associated with a set of rules as criteria (Wang & Rudin, 2014; Chen & Rudin, 2017). While prototype could be selected from actual patients and genes for clinicians to make sense of large patient cohort or gene data (Bien & Tibshirani, 2011). However, there are still open challenges: How to construct simple rules with more accurate prediction and classification performance? How to produce accurate and intuitive definitions of prototypes?

In this work, we propose Prototype lEArning via Rule List (PEARL), which combines rule learning and prototype learning on deep neural networks to harness the benefits of both approaches and alleviate their shortcomings for an accurate and interpretable prediction model. In particular, we iteratively learn rule lists, via a data reweighing procedure using prototypes, and then update prototypes via neural networks with learned rules. PEARL not only generates simple and interpretable rule lists and prototypes, but also provides neural network models which can infer the similarity of a query datum to all the prototypes. To summarize, we make the following contributions in this paper.

1. We propose an integrative method to combine rule list and prototype learning, enabling PEARL to harness the power of these methods.

2. PEARL automatically learns prototypes corresponding to rules in a rule list, which are more concise than conventional rule list learning methods and more explainable than prototype learning methods by providing logic reasoning.

3. On real-world electronic health record datasets, PEARL demonstrates both accurate prediction performance and simple interpretation.

## 2 TECHNICAL BACKGROUND

A prototype is an object that is representative of a set of similar instances (e.g., a patient from a cohort) and can be a part of the observed data points or an artifact summarizing a subset of them with similar characteristics. Prototype learning is a type of case-based reasoning (Kolodner, 1992) and aims to find some prototypes (Priebe et al., 2003; Bien & Tibshirani, 2011; Kim et al., 2014) Prototypes can be seen as an alternative approach to learn centroids of clusters, and have been applied to few shot learning (Mensink et al., 2013; Rippel et al., 2015; Vinyals et al., 2016; Snell et al., 2017). Let $X = \{x_i\}_{i=1}^n$ be the data set, to learn one prototype $p_j$ of many prototypes, existing works choose some $p_j \in X$ (Bien & Tibshirani, 2011), compute a linear combination $p_j = \sum_i^n b_{ij} x_i$ (Wu & Tabak, 2017), or form a Bayesian generative model (Kim et al., 2014). In this work, we follow (Li et al., 2017) and use a general representation of $p_j = f_j(X)$, where $f_j$ is automatically learned via deep neural networks. $p_j$ has the same dimension as the learned representation of data, which is a predefined hyperparameter. Current prototype selection methods typically select one prototype at a time, and provide limited higher-level abstraction on the reasoning side of diagnosis.

Rule lists are logic statements over original features. A rule list $R = (r_1, r_2, ..., r_K, r_0)$ of length $K$ is a $(K+1)$-tuple consisting of $K$ distinct association rules, $r_k := z_k \Rightarrow q_k$ for $k = 1, ..., K$ with an additional default rule $r_0$. Each rule $r = z \to q$ is an implication corresponding to the conditional statement, "if $z$, then $q$" where $z$ is premise and $q$ is conclusion. In general, rule lists are easy to understand. In this paper, each $r_i$ in $R$ are dependent of previous rules with "else-if" logics. We build on existing rule list learning method (Angelino et al., 2018) to iteratively guide the prototype learning via neural networks. In addition, we use rule learning methods where each individual rule consists of logic AND clauses but not ORs.

Following existing definition of interpretability (Lakkaraju et al., 2016), there are 4 aspects of interpretability: size, length, cover, and overlap.

**Size**. The size of a rule lists is defined as number of rules $K$ in a rule list $R$. The fewer the rules in a rule list, the easier it is for a user to understand all of the conditions that correspond to a particular class.

**Length.** We use the term length to measure the number of clauses in each rule $r_i$. If the number of clauses in a rule is too large, it will loose its natural interpretability.

**Cover.** Cover measures the set of data points that satisfy each $r_i$. Cover measures how the data is divided by the rule classifiers.

**Overlap.** Overlap between two rule $r_i$ and $r_j$ is the number of points that satisfy both rules. It measures the discriminative power of each rule and whether decision boundary is clearly defined.

In this paper, we mainly investigate and provide new methods to reduce the size (by combining rule in rule lists into prototypes) and the length (by replacing the clauses in each rule using a prototype) of rule list classifiers, while improving on the accuracy of rule lists. We propose to use the cover of each rule $r_i$ to re-weight data, which forces rule learning methods to focus on more discriminative data points and hence reduce the overlap among rules.

## 3 PEARL: METHODOLOGY

Let $\mathcal{X} = \{X_1, \cdots, X_N\}$ be $N$ data samples, where each sample $X_n$ (e.g., health records for patient $n$) is a sequences of discrete event labels (such as medical codes in electronic health records). We can represent $X_n$ as $\{e_n^i, t_n^i\}$, where $e_n^i \in \mathcal{E}$ is the $i$-th event label in $X_n$ and $t_n^i$ is the time stamp of $e_n^i$. For each $X_n$, there is a class label $y_n$. For example, in health applications, $\boldsymbol{y}$ are the classification result of targeting diseases such as the onset of heart failure (binary), or subtypes of diabetes (multiclass). The goal of PEARL is to accurately predict $\boldsymbol{y} = \{y_1, \cdots, y_N\}$ and to provide explanation for such predictions. In this work, we assume both $X_n$ and $y_n$ are categorical variables.

In this work, we aim to do so by providing an interpretable representation of data with a deep neural network. The outputs of the network include the class label $y$ and a set of interpretable prototypes $\hat{P}$ corresponding to a rule list $R$. The neural network is used to performing accurate classification, under the guidance of prototypes defined by the rule lists. Formally, the overall objective of PEARL is:

$$\underset{\theta_1,\theta_2}{\arg\min} \underbrace{\lambda_1 \mathcal{L}_1(\mathbf{h}(\mathcal{X};\theta_1), \hat{P})}_{\text{distance of data to prototypes}} + \underbrace{\lambda_2 \mathcal{L}_2(\mathbf{s}_R(\mathbf{h}(\mathcal{X};\theta_1);\theta_2), \boldsymbol{y})}_{\text{classification error}},$$

$$\text{where} \quad \mathcal{L}_1(\mathbf{h}(\mathcal{X};\theta_1), \hat{P}) = \sum_{X_n \in \mathcal{X}} \min_{k \in \{1,\cdots,K\}} d(\mathbf{h}(X_n;\theta_1), \mathbf{p}_k), \quad (1)$$

$$\text{and} \quad \hat{P} = f(\mathbf{h}(\mathcal{X};\theta_1), R). \quad |\hat{P}| \leq |R|.$$

where $\boldsymbol{h}(\mathcal{X};\theta_1)$ is the learned representation of $\mathcal{X}$ with parameter $\theta_1$. $\boldsymbol{h}(\mathcal{X};\theta_1)$ is a vector and has the same predefined dimension as $p_k$, $R$ is the learned rule list, and $P$ is the set of learned prototypes. A set of prototypes $\hat{P} = \{\mathbf{p}_1, \mathbf{p}_2, \cdots, \mathbf{p}_K\}$ contains $K$ representation of data, which serve as prototypes. Here $d$ is a distance measure, such as the cosine distance. $f$ is a fixed mapping that, given $R$ and learned $\boldsymbol{h}(\mathcal{X};\theta_1)$, $\hat{P}$ are determined without further learning. More details on $f$ can be found in Section 3.1.2. Each $\mathbf{p}_k$ lies in the same space as $\mathbf{h}(\mathcal{X};\theta_1)$, and should correspond to one or more rules in $R$. The second term $\mathcal{L}_2$ is the Cross Entropy loss for the final prediction target, where $\mathbf{s}_R(\mathbf{h}(\mathcal{X};\theta_1);\theta_2)$ represents the predicted label for $\mathcal{X}$ and $\boldsymbol{y}$ is the ground-truth label. Here $\theta_1$ represents all the model parameters for data representation learning $\mathbf{h}(\mathcal{X};\theta_1)$ and $\theta_2$ represents those of classification model $\mathbf{s}_R(\cdot)$. We will drop $\theta$s for simplicity from now on. Minimizing $\mathcal{L}_1$ would encourage training examples to be as close as possible to at least one prototype in the latent space, motivated by (Li et al., 2017). However, we do not use other terms from (Li et al., 2017) and instead introduce rule lists as the guidance for prototype learning. Note that relative weights $\lambda_1$ and $\lambda_2$ values are chosen via hyperparameter tuning. In general we chose $\lambda_2 > \lambda_1$ to emphasize the classification performance.

Since it is non-trivial to integrate rule and neural network learning, we propose a framework, PEARL, of integrating rule learning and rule-guided prototype learning together. The main intuition is to learn and produce prototypes that are closely related to rules in $R$, with one-to-one or many-to-one rule-prototype mapping. This serves as a constraint to make each prototype as a surrogate for clauses in each rule, transforming "if data $x$ satisfies $z$, then $x = q$" to "if $x$ is close to a prototype $p$, then $x = q$". We will discuss the network structures in details next.

Table 1: Notations used in `PEARL`.

| Notation | Definition |
|---|---|
| $E; e_n^i \in \{1, 2, \cdots |E|\}$ | All events; Event $i$ of subject $n$ |
| $t^i | i \in \{1, \cdots, T\}$ | Time stamp for event $i$ |
| $R = (r_1, r_2, \cdots, r_K, r_0)$ | Rule list comprised of $K$ rules, $r_0$ is the default rule |
| $X_n = \{e_n^1, t_n^1; e_n^2, t_n^2; \cdots\}$ | Event sequence of subject $n$ |
| $\boldsymbol{y}; y_n$ | Labels for all data $\mathcal{X}$; One label for sequence $X_n$ |
| $\mathcal{L}_1; \mathcal{L}_2$ | Loss for prototype similarity; Cross-entropy loss for classification |
| $\hat{P} = \{\mathbf{p}_1, \mathbf{p}_2, \cdots, \mathbf{p}_K\}, \mathbf{p}_i \in \mathbb{R}^c$ | $K$ prototype vectors, prototype layer in network. |
| $\mathbf{h}(X) \in \mathbb{R}^c; \mathbf{s}_R(X)$ | Output of highway layer; Output of softmax layer |
| $\mathbf{o}_R(X) \in \mathbb{R}^K$ | Output of prototype layer, subscript $R$ mean it rely on rule list. |
| $r_i \rightarrow \mathbf{p}_i$ | One prototype $\mathbf{p}_i$ corresponds to a rule $r_i$ |
| $\mathcal{X}; \mathcal{X}^{(j)}$ | Training subjects; Training subjects that satisfy rule $r_j$ |

## 3.1 MODEL ARCHITECTURE

The network architecture of `PEARL`, illustrated in Fig. 1, mainly comprises two modules: an interpretable module with a rule list learning procedure, and a prediction module with a prototype learning procedure.

The interpretable module generates a rule list given input data $X$ and pass it to the prediction module. The prediction module consists of a representation network and a prototype learning network. The representation network is made of a temporal modeling component, followed by a highway network with skip connections to alleviate the numerical issue of vanishing gradients (He et al., 2016). The prototype learning network learns prototypes based on the rules from interpretation module and learned representation from the representation network, and then uses prototypes for the final prediction. Moreover, the prediction module also re-weights data per distance to learned prototypes. The re-weighted data is then used again for learning a new rule list and new data representation.

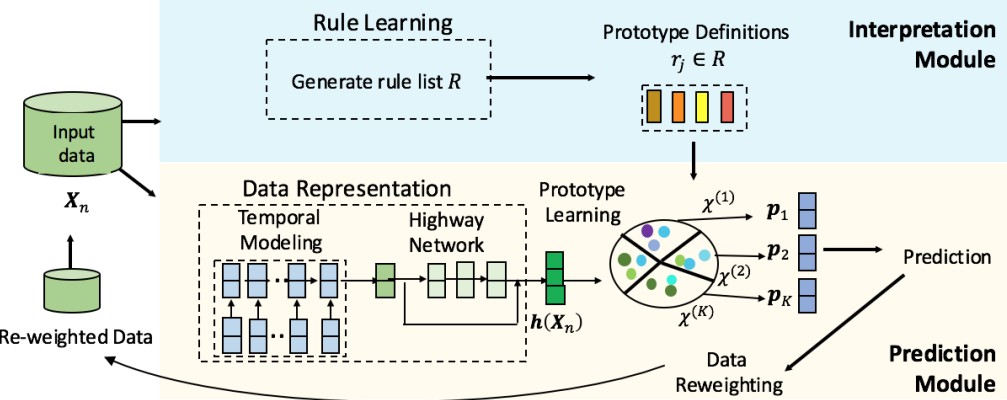

Figure 1: The `PEARL` architecture includes two modules: an interpretable module with a rule list learning procedure, and a prediction module with a prototype learning procedure. Two modules iteratively improves each other during training.

Overall, the prediction module iteratively uses rule lists to guide the prototype learning via a neural network. Then the interpretation module iteratively re-weights the data and updates its own rule learning. The two modules are discussed in more details below.

### 3.1.1 INTERPRETATION MODULE: RULE LEARNING

The interpretable module employs a rule list classifier to provide interpretable prototype definitions. Given data $X_n$, we use a known rule list learning algorithm to generate a rule list $R$, with size

$|R|$. In general, any rule list algorithm can be adopted, and we choose one recent state-of-the-art COREL (Angelino et al., 2018). $R$ is then used to help the prediction module to define and interpret prototypes. If the feature size is too large, we can apply a feature selection algorithm to reduce feature dimension. We tested a few feature selection algorithms in our experiments and they do not impact the performance much, if any at all. We will discuss prediction module next and then discuss how interpretation module can benefit from the prediction module in an iterative data re-weighting procedure.

### 3.1.2 PREDICTION MODULE: NEURAL NETWORK

The prediction module contains a patient representation learning and a prototype learning network.

**Data Representation via Neural Networks** To encode patient longitudinal clinical events, we first embed the event sequences using neural networks. Although we have flexible choices of neural networks, in this paper we chose the recurrent convolution neural networks (RCNN) (Liang & Hu, 2015) to learn the distributed representations of each event. In particular, we added one dimension filter and a max-pooling layer in the CNN part, and used a bidirectional LSTM for RNN. This representation learning procedure for patient $n$ is denoted as Eq. 2.

$$\mathbf{g}_n = \text{RCNN}(X_n) = \text{RCNN}([e_n^1, \tau_n^1], [e_n^2, \tau_n^2], \cdots, [e_n^t, \tau_n^t]), \tag{2}$$

where $\tau_n^k$ is the time difference between consecutive events, such that $\tau_n^k = t_n^k - t_n^{k-1}$ for $k > 1$ and $\tau_n^0 = 0$. By including $\tau_n^k$ as additional features, we incorporate the time information into patient representation learning. After RCNN we also use highway network (Srivastava et al., 2015) to alleviate the vanishing gradient issue in network training. A single layer of highway network is:

$$\mathbf{y} = H(\mathbf{x}, \mathbf{W}_H) \odot T(\mathbf{x}, \mathbf{W}_T) + \mathbf{x} \odot (1 - T(\mathbf{x}, \mathbf{W}_T)), \tag{3}$$

where $\mathbf{x}$ and $\mathbf{y}$ are input and output for a single layer, respectively. Here $\odot$ is element-wise multiplication, $T$ is the transform gate, and the dimensionality of $\mathbf{x}, \mathbf{y}, H(\mathbf{x}, \mathbf{W}_H)$, and $T(\mathbf{x}, \mathbf{W}_T)$ are the same. $T$ and $H$ use sigmoid and Relu as activation function, respectively. Multiple layers highway network are concatenated. Given $\mathbf{g}_n$ as input of the first layer of highway networks, after multiple layers of updating, we represent the output of the $n$-th sample as $\mathbf{h}(X_n)$, which can be simplified as

$$\mathbf{h}(X_n) = \text{Highway-Network}(\mathbf{g}_n). \tag{4}$$

Empirically we find the highway networks are essential for prototype qualities.

Data representation learning step is not limited to the combination or RCNN and highway network. To generalize this representation learning step, we can write $\mathbf{h}(X_n) = \text{Encoder-NN}(X_n)$, which is the composite of Equation 2 and 4.

**Rule-guided Prototype Learning** The embedded clinical events $\mathbf{h}(X_n)$ is then used in an iterative prototype learning procedure. Specifically, we first generate prototype vectors from $\mathbf{h}(X_n)$. Given a rule list $R$, $|R| = K$, for each rule $r_j \in R$, we can find all positive data samples for $r_j$, denoted as $\mathcal{X}^{(j)}$. Thus we can get a pseudo representation of $r_j$:

$$\mathbf{p}_j = \frac{1}{|\mathcal{X}^{(j)}|} \sum_{X_i \in \mathcal{X}^{(j)}} \mathbf{h}(X_i), \quad \text{for } j = 1, \cdots, K. \tag{5}$$

where $\mathcal{X}^{(j)} \subseteq \mathcal{X}$ represent all the data samples that satisfy the $j$-th rule $r_j$. $|X^{(j)}|$ represents its cardinality. The output of prototype learning network is a vector of one training subject's distance to all the prototypes, as given by Eq. 6.

$$\mathbf{o}_R(X_n) = \left[ \text{d}(\mathbf{h}(X_n), \mathbf{p}_1), \text{d}(\mathbf{h}(X_n), \mathbf{p}_2), \cdots, \text{d}(\mathbf{h}(X_n), \mathbf{p}_K) \right]^\top \in \mathbb{R}^K. \tag{6}$$

Here, $\text{d}(\mathbf{v}_1, \mathbf{v}_2) = \frac{\mathbf{v}_1^\top \mathbf{v}_2}{\|\mathbf{v}_1\|_2 \cdot \|\mathbf{v}_2\|_2}$, is the cosine distance of $\mathbf{v}_1$ and $\mathbf{v}_2$. The dimension of $\mathbf{o}(X)$ depends on the number of rules. Since these prototypes use rules as guidance, we also call them rule-prototypes, in contract to non-rule prototypes in (Li et al., 2017). The subscript $R$ means the function rely on rule list $R$.

Last, a fully-connected layer (with parameter $\mathbf{W} \in \mathbb{R}^{K \times L}$, where $L$ is number of class) and a softmax activation are used to perform the final classification.

$$\mathbf{s}_R(X_n) = \text{softmax}(\mathbf{W}^\top \mathbf{o}(X_n)), \tag{7}$$

where $\mathbf{s}_R(X_n)$ is the estimated probability. We then used the standard cross-entropy loss for training.

### 3.1.3 Iterative Data Reweighing

To enable the iterative learning of prototypes and rule list, we use a data re-weighting procedure based on results from the prediction module. We first provide some intuition and then describe the detailed method.

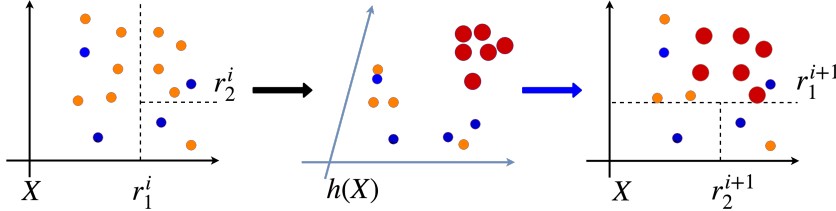

Figure 2: Illustration of the process in which prototypes and rule list learning affect each other via up-weighting more discriminative samples (shown as bigger red dots). Best viewed in color.

**Intuition** Since learned prototypes are trained to represent spatially close data samples from the new learned feature space $\mathbf{h}(X)$, prototypes can be more discriminative and can reveal more of the underlying data similarity relationships than the rules from the original feature space as shown in the 2nd diagram of Figure 2. With such a better similarity measure from the representation space, new representations of data samples are more easily separable. More importantly, the examples that are difficult to separate may often be noises or low probability examples, i.e., if $p(x, y)$ be the joint distribution of data, a hard-separable example $x_i$ has low $p(x_i, y_i)$. Such a phenomenon has been observed previously in training simpler models (Dhurandhar et al., 2018). If we up-weight simpler samples that are more separable, rule-list learning focuses these simpler samples more and lead to easier training and more separation later. For examples, the red dots shown in Figure 2 are the high-probability examples, which should be given higher weights. We will also empirically study data separation in experiments to justify this intuition.

**Procedure** The iterative learning and re-weighing procedure is based on the similarity between each data sample (such as patient subjects) and prototypes. To start with, we measure the cosine similarity between subject $\mathbf{h}(X_n)$ and each prototype vector $\mathbf{p}_k$ as depicted by Eq. 8.

$$s_{nk} = d(\mathbf{h}(X_n), \mathbf{p}_k), \tag{8}$$

where d is cosine distance measure. We aim at boosting the prototypes that have fewer subjects within its proximal neighbors in the learned representation space, indicating these prototypes are far away from other subjects and hence more discriminative. Thus, for each prototype $k$, we calculate its average similarity with all subjects as $s^k = \frac{1}{N} \sum_{n=1}^{N} s_{nk}$, where $N$ is the size of the current dataset. Then we collect those prototypes, denoted as $\mathcal{K}'$, of which $s^k$ is less than a pre-specified threshold $\eta$ and their corresponding data subjects. We concatenate these samples to the original data $X$.

$$\mathcal{X} \leftarrow [\mathcal{X}, \mathcal{X}^{(j)}] \ \forall j \in \mathcal{K}', \tag{9}$$

where $\mathcal{X}^{(j)} \subseteq \mathcal{X}$ represent all the data samples that satisfy the $j$-th rule. We summarize the procedure in Algorithm 1. We alternately optimize rule list $R$ and neural networks until convergence. The convergence criteria is when the loss of the current epoch is within a pre-specified threshold from the previous epoch. Data augmentation is equivalent to data weighting. For practical purposes where the rule list cannot directly handle data weights, data augmentation can achieve desired results.

**Inference Procedure for New Samples** For a new subject $X_{\text{new}} = \{e_{\text{new}}^1, e_{\text{new}}^2, \cdots, \}$, PEARL will generate two outputs. First is the predicted probability for classification, i.e., the output in softmax layer, $\mathbf{s}_R(X_{\text{new}})$ in Eq. 7. Second, we obtain the output of prototype layer, i.e., $o(X_{\text{new}})$. As it indicate the similarity between the current example and prototypes by their cosine distance, the new subject can be explained by the characteristics of its closest prototype.

---

**Algorithm 1** `PEARL` Prototype Learning via Rule lists

---

**Input:** Event sequence $X_n = \{e_n^1, t_n^1; e_n^2, t_n^2; \cdots, \}$, label $y_n$, corresponding binary vector $\mathbf{c}_n = [0, 0, 1, \cdots]^\top$ for $n = 1, \cdots, N$. Let $N = |\mathcal{X}|$. Hyperparameter $\eta$, maximal iteration number $T_{\max}$.
**Output:** rule list, network classifier.

1: **for** iter $= 1, \cdots, T_{\max}$ **do**
2:     **A. Rule Learning**:
3:       If needed, apply feature selection to reduce feature dimensions. $\mathbf{c}_n$ is transformed into low-dimensional $\tilde{\mathbf{c}}_n = [0, 1, \cdots]^\top$.
4:       Find rule $R = \{r_1, r_2, \cdots, \}$ based on $\tilde{\mathbf{c}}$. $\mathcal{X}^{(j)} \subseteq \mathcal{X}$ is set of all samples fit the rule $r_j$.
5:     **B. prototype + NN training**: Construct and train the neural network (Section 3.1).
6:     **C. Data Reweighing**:
7:       Compute all $s_{nk}$, i.e., similarity between $n$-th data and $k$-th prototype.
8:       Collect all prototypes $k \in \{1, \cdots, K\}$ that have less corresponding subjects ($s^k = \frac{1}{N} \sum_n s_{nk} < \eta$) into $\mathcal{K}'$.
9:       Reweigh data according to Eq. 9: $\mathcal{X} \leftarrow [\mathcal{X}, \mathcal{X}^{(j)}] \ \forall j \in \mathcal{K}'$. $N = |\mathcal{X}|$.
10: **end for**

---

Table 2: Basic statistics of datasets

| Dataset | Heart Failure Dataset | MIMIC-III (mortality prediction) |
|---|---|---|
| # cases | 2,268 | 2,825 |
| # controls | 14,526 | 4,712 |
| # visits per patient | 19.7 | 2.7 |
| # clinical variables per patient | 41.0 | 21.6 |
| # unique clinical variables | 1,865 | 942 |
| # clinical variables per visit | 2.1 | 11.6 |

## 4 EXPERIMENT

### 4.1 EXPERIMENTAL SETUP

We evaluate `PEARL` model by comparing against other baselines on two tasks: heart failure (HF) detection and mortality prediction. All methods are implemented in PyTorch (Paszke et al., 2017) and trained on a laptop with 8GB memory.

**Dataset Description** To evaluate the performance of `PEARL`, we conducted experiments using the following real world datasets. The statistics of the datasets are summarized in Table 2.

Heart Failure (HF) Data: The HF dataset is extracted from a proprietary EHR warehouse [1] where subjects were generally monitored over 4 years. The HF cohort includes $2,268$ case patients and $14,526$ matching controls as defined by clinical experts. Subject inclusion criteria is in Appendix.

MIMIC-III Data: We use the MIMIC III (Medical Information Mart for Intensive Care) data for evaluation[2]. MIMIC III is collected on over 58,000 ICU patients at the Beth Israel Deaconess Medical Center from June 2001 to October 2012 (Johnson et al., 2016). We only included patients with at least two visits in our experiment, resulting in a total of 7,537 ICU patients.

**Baselines** We consider the following baseline algorithms.

- Rule learning: in this work we used the certifiably optimal rule lists in Angelino et al. (2018).

- Decision Tree: we directly use scikit (Pedregosa et al., 2011) package in Python.

- Prototype Learning (without rules) (Li et al., 2017): RCNN+prototype (without rule). Prototype is randomly initialized. The result is very sensitive to the initialization.

- RNN (Doctor-AI) (Choi et al., 2016a): RNN+softmax. It concatenate multi-hot vector with a difference of time stamp as input feature. A softmax layer is added after bi-LSTM.

---

[1] Data source is anonymized for blind review.
[2] `https://mimic.physionet.org/`

- RCNN: CNN+RNN+softmax. All RCNN use 1 dimensional filter, a max-pool layer and bi-LSTM. It is followed by a softmax layer.

**Evaluation Strategies** We randomly split dataset 5 times and repeat the experiments with different random seeds. For each split, we divide the dataset into training, validation and testing set in a $7:1:2$ ratio. Then we report the mean and standard deviation of results (both accuracy and run time). To measure the prediction accuracy, we used the area under the receiver operating characteristic curve (ROC-AUC). For rule learning, we report the average results of 5 trials. After tuning, we set $\lambda_1 = 1$ and $\lambda_2 = 1e-3$. To initialize embeddings, we use window size of 15 for word2vec (Mikolov et al., 2013) and train medical code vectors of 100 dimensions on each training data, following (Ma et al., 2018). For prototype learning, we use the same number of prototypes with PEARL to make sure that the parameter numbers are the same. For the RNN model, we implemented a bidirectional-LSTM with hidden layer size 3. For the RCNN model, the number of filters for CNN is 30, stride is 1, and the windows size is 1. We add a max-pooling layer following convolution with pool size $(5, 1)$. For the highway network, the number of layers of highway network is set to 2. Training is done through Adam (Kingma & Ba, 2014) at learning rate 1e-1. The batch size is set to 256. Data weighting threshold $\eta$ is set to values between .45 and .55. The threshold in convergence criteria is set as 0.001. We fix the best model on the validation set within 5 epochs and report the performance in the test set.

## 4.2 RESULTS

**Accuracy Comparison** Table 3 shows PEARL has the highest AUC performance among all methods. As for the baseline models, the rule learning has the lowest AUC due to it makes classification based on composition of simple logics. Prototype learning is better than rule learning and RNN models but worse than PEARL. It shows PEARL can improve upon both prototype and rule learning.

Table 3: Performance Comparison of Different Methods. Runtime is measured in terms of seconds. The numbers in parenthesis are the standard deviation.

| Model | Heart Failure | | | MIMIC-III | | |
|---|---|---|---|---|---|---|
| | ROC-AUC | Runtime | # param | ROC-AUC | Runtime | # param |
| Rule List(Angelino et al., 2018) | .533(.001) | 78.8(.5) | 0.1K | .485(.015) | 47.5(.7) | 0.07K |
| Decision Tree | .530(.003) | 2.87(.3) | 0.4K | 0.657(0.008) | 0.92(0.09) | 0.6K |
| Prototype Learning (Li et al., 2017) | .668(.003) | 768.3(6.3) | 18.4K | .761(.010) | 178.0(5.7) | 18.4K |
| RNN (Choi et al., 2016a) | .636(.008) | 1405.0(27.0) | 95.8K | .724(.011) | 318.0(11.2) | 49.7K |
| RCNN | .682(.009) | 718.0(13.7) | 8.4K | .766(.009) | 144.0(3.4) | 8.4K |
| PEARL with 1 epoch | .673(.009) | 783(67) | 18.4K | .761(.006) | 237(8.9) | 18.4K |
| PEARL | **.683(.003)** | 1538(73) | | **.768(.009)** | 318.1(47.2) | |

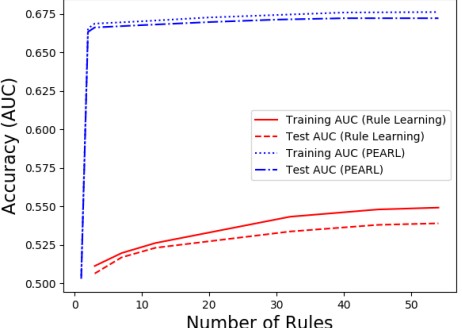

Figure 3: Tradeoff: AUC v.s. Interpretability. PEARL is more robust to number of rules. Conventional rule learning method may require lots of rule and generalize poorly on test dataset.

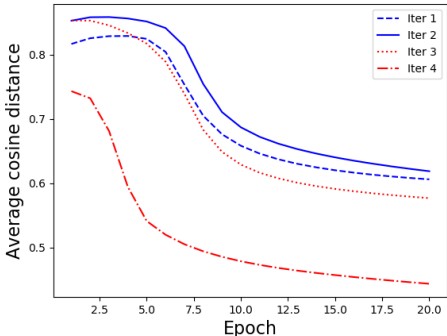

Figure 4: Average Distance between data and prototype during training iterations.

**Interpretability-Accuracy Tradeoff** We study the relationship between accuracy and the interpretability in rule list learning and the proposed `PEARL` model. Interpretability is measured by the number of rules (also the number of prototypes) of different methods. Figure 3 shows that our method can use a small number of prototypes to achieve better accuracy than the rule list learning. In fact, 3 rule-prototypes can already explain more samples than rule lists with over 50 rules.

**Separating distance of Prototypes over Training Iterations** To justify reweighing data points lead to more separable prototypes, we study on whether the mean distances $\frac{\sum_{k=1}^{K}\sum_{n=1}^{N} s_{nk}}{NK}$ between prototypes and data decreases over training iteration $T_i$. As shown in Figure 4, by using data weighing, the average distance decreases across iteration $T_i$. Interestingly, even with each iteration, the average distance also decreases with training epochs of neural networks, suggesting that such a reduction in average distances also leads to lower training loss.

We conducted further experiments to test the accuracy change of rule lists during training, which shows that data augmentation helps improving rule list accuracy, along with more hyper-parameter tuning results. All the results are shown in the appendix.

### 4.3 CASE STUDY

We study whether `PEARL` can provide more interpretable diagnosis compared with conventional rule learning. In particular, we find the corresponding prototypes learned in `PEARL` for a sets of patients and retrieve the closest rule-prototypes. For each prototype with multiple patients, we retrieve their high frequent events among the patients who satisfy the rule-prototype while the remaining events that only occur to one or two patients are discarded.

In general, the rule learning often yields complex rule lists that involve hundreds of clinical events, many of which are duplicated in multiple rules. As a contrast, `PEARL` only used $\sim 10$ rules to make correct diagnosis. Below we provide one example of prototype-rules from `PEARL`.

```
If a patient experience all following events:
(1) chronic airways obstruction, (2) malignant neoplasm of trachea, lung
 and bronchus, (3) carcinoma in situ of respiratory system, (4) Alprazolam,
(5) Eszopiclone, (6) abnormal findings on radiological examination of body
structure, (7) acute bronchitis, (8) Albuterol Sulfate,
(9) Hypertrophic conditions of skin, and (10) diltiazem hydrochloride,
then the patient has a high probability of experiencing heart failure.
```

The prototype-rules include 10 clinical events. Most of them concern severe conditions of lung and respiratory systems (a common symptom of HF patients), and the medications for treating HF, which are common comorbidities of heart failures. Patients belong to this prototype can be diagnosed based on the occurrence of these events on their EHR. For patients of this prototype, if using conventional rule learning, diagnosis would require a much more complex rule with more than 40 clinical events and rule depth for about 50. We provide one example in A.2 in Appendix.

## 5 CONCLUSION

In this paper, we proposed `PEARL`, an integrative prototype learning neural network that combines rule learning and prototype learning on deep neural networks to harness the benefits of these methods. We empirically demonstrated that `PEARL` is more accurate , thanks to an iterative data reweighing algorithm, and more interpretable than rule learning, since it explains diagnostic decisions using much fewer clinical variables. `PEARL` is an initial attempt to combine traditional rule learning with deep neural networks. In future research, we will try to extend `PEARL` to other interpretable models.

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

## A  DETAILS OF DATASETS

### A.1  INCLUSION CRITERIA FOR HEART FAILURE DATA

The criteria for being patients include 1) ICD-9 diagnosis of heart failure appeared in the EHR for two outpatient encounters, indicating consistency in clinical assessment, and 2) At least one medication was prescribed with an associated ICD-9 diagnosis of heart failure. The diagnosis date was defined as its first appearance in the record. These criteria have also been previously validated as part of Geisinger Clinical involvement in a Centers for Medicare and Medicaid Services (CMS) pay-for-performance pilot (Pfisterer et al., 2009). For matching controls, a primary care patient was eligible as a control patient if they are not in the case list, and had the same gender and age (within 5 years) and the same PCP as the case patient. More details could be found in (Wu et al., 2010).

### A.2  EXAMPLE RESULT OF RULE LEARNING

For the diagnosis of patients in the prototype mentioned in Sec.4.3, we need to check all rules below.

```
if (RETINAL DISORDERS=yes,NONSUPPURATIVE OTITIS MEDIA AND EUSTACHIAN TUBE DISORDERS=yes) then (Yes)

else if (WARFARIN SODIUM=yes, CONDUCTION DISORDERS=yes) then (Yes)

else if (DILTIAZEM HYDROCHLORIDE=yes, PULMONARY CONGESTION AND HYPOSTASIS=yes) then (Yes)

else if (AMIODARONE HYDROCHLORIDE=yes) then (Yes)

else if (DIGOXIN=yes,OTHER AND UNSPECIFIED ANEMIAS=yes) then (Yes)

else if (ATHEROSCLEROSIS=yes,CODEINE PHOSPHATE=yes) then (Yes)

else if (ABNORMAL FINDINGS ON EXAMINATION OF BODY STRUCTURE=yes, DIGOXIN=yes) then (Yes)

else if (ASSAULT BY SUBMERSION [DROWNING]=yes,CONDUCTION DISORDERS=yes) then (Yes)

else if (OTHER AND UNSPECIFIED ANEMIAS=yes,CODEINE PHOSPHATE=yes) then (Yes)

else if (ALBUTEROL SULFATE=yes,THEOPHYLLINE ANHYDROUS=yes) then (Yes)

else if (ATHEROSCLEROSIS=yes,CONDUCTION DISORDERS=yes) then (Yes)

else if (WARFARIN SODIUM=yes,ALPRAZOLAM=yes) then (Yes)

else if (ANGINA PECTORIS=yes,ALPRAZOLAM=yes}) then (Yes)

else if (ACUTE BRONCHITIS AND BRONCHIOLITIS=yes,PULMONARY CONGESTION AND HYPOSTASIS=yes) then (Yes)

else if (WARFARIN SODIUM=yes,THEOPHYLLINE ANHYDROUS=yes) then (Yes)

else if (HEARING LOSS=yes,CODEINE PHOSPHATE=yes) then (Yes)

else if (DISEASES OF PERICARDIUM=yes,OTHER BACTERIAL PNEUMONIA=yes) then (Yes)

else if (DISEASES DUE TO VIRUSES AND CHLAMYDIAE=yes,ANGINA PECTORIS=yes}) then (Yes)

else if (DIABETES MELLITUS=yes,PHYSIOLOGICAL MALFUNCTION ARISING FROM MENTAL FACTORS=yes) then (Yes)

else if (CHRONIC AIRWAYS OBSTRUCTION=yes,CODEINE PHOSPHATE=yes) then (Yes)

else if (DISEASES OF PERICARDIUM=yes,OTHER BACTERIAL PNEUMONIA=yes) then (Yes)

else if (DISEASES DUE TO VIRUSES AND CHLAMYDIAE=yes,ANGINA PECTORIS=yes) then (Yes)

else if (DIABETES MELLITUS=yes,PHYSIOLOGICAL MALFUNCTION ARISING FROM MENTAL FACTORS=yes) then (Yes)

else if (CHRONIC AIRWAYS OBSTRUCTION=yes,CODEINE PHOSPHATE=yes) then (Yes)

else if (ALPRAZOLAM=yes,CONDUCTION DISORDERS=yes) then (Yes)

else if (ANGINA PECTORIS=yes,ESZOPICLONE=yes) then (Yes)

else if (DISORDERS OF REFRACTION AND ACCOMMODATION=yes,DISORDERS OF PLASMA PROTEIN METABOLISM=yes) then (Yes)

else if (OLD MYOCARDIAL INFARCTION=yes,MALIGNANT NEOPLASM OF TRACHEA, BRONCHUS, AND LUNG=yes}) then (Yes)

else if (NEOPLASM OF UNCERTAIN BEHAVIOR OF OTHER AND UNSPECIFIED SITES AND TISSUES=yes,OTHER BACTERIAL PNEUMONIA=yes) then (Yes)

else if (OTHER HYPERTROPHIC AND ATROPHIC CONDITIONS OF SKIN=yes,OTHER BACTERIAL PNEUMONIA=yes) then (Yes)

else if (VISUAL DISTURBANCES=yes,PULMONARY CONGESTION AND HYPOSTASIS=yes) then (Yes)

else if (HYDROCORTISONE=yes,CODEINE PHOSPHATE=yes) then (Yes)

else if (NEOMYCIN SULFATE=yes,OTHER HYPERTROPHIC AND ATROPHIC CONDITIONS OF SKIN=yes) then (Yes)
```

```
else if (RETINAL DISORDERS=yes,ALPRAZOLAM=yes) then (Yes)

else if (DIABETES MELLITUS=yes,OTHER POSTSURGICAL STATES=yes) then (Yes)

else if (ANGINA PECTORIS=yes,CONDUCTION DISORDERS=yes) then (Yes)

else if (ANGINA PECTORIS=yes,ATHEROSCLEROSIS=yes) then (Yes)

else if (OTHER DISORDERS OF KIDNEY AND URETER=yes,PULMONARY CONGESTION AND HYPOSTASIS=yes) then (Yes)

else if (OTHER AND UNSPECIFIED ANEMIAS=yes,CONDUCTION DISORDERS=yes) then (Yes)

else if (MALIGNANT NEOPLASM OF TRACHEA, BRONCHUS, AND LUNG=yes,CONDUCTION DISORDERS=yes) then (Yes)

else if (ANGINA PECTORIS=yes,OTHER DISEASES OF PERICARDIUM=yes) then (Yes)

else if (CODEINE PHOSPHATE=yes) then (Yes)

else if (OTHER RETINAL DISORDERS=yes,PHYSIOLOGICAL MALFUNCTION ARISING FROM MENTAL FACTORS=yes) then (Yes)

else if (ESZOPICLONE=yes) then (Yes)

else if (DIABETES MELLITUS=yes,SPECIAL INVESTIGATIONS AND EXAMINATIONS=yes) then (Yes)

else if (DIABETES MELLITUS=yes,OTHER AND UNSPECIFIED DISORDERS OF BACK=yes) then (Yes)

else if (HEARING LOSS=yes,OTHER DERMATOSES=yes) then (Yes)

else if (DISORDERS OF REFRACTION AND ACCOMMODATION=yes,OTHER AND UNSPECIFIED DISORDERS OF BACK=yes) then (Yes)

else if (MALIGNANT NEOPLASM OF TRACHEA, BRONCHUS, AND LUNG=yes,ESZOPICLONE=yes) then (Yes)

else if (DILTIAZEM HYDROCHLORIDE=yes,DIABETES MELLITUS=yes) then (Yes)

else (No)
```

## A.3 RULE LEARNING ACCURACY AS A FUNCTION OF EPOCHS

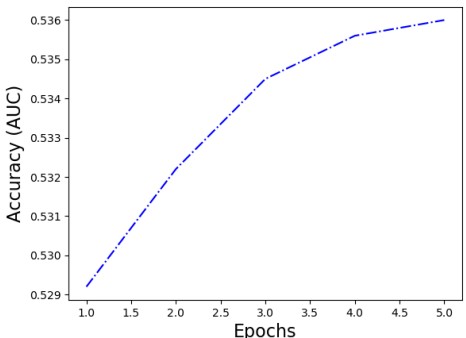

Figure 5: Average rule accuracy on test dataset for different epochs

Here, we study the accuracy of rule during different epochs in Algorithm 1. We conduct 5 independent trials using different hyperparameter and report their average results. The results are shown in Figure 5. We can find that the accuracy of rules increase with iterative learning and we conclude that the data augmentation does improve the accuracy of rule list learning as well.

## A.4 RULE PROTOTYPE VS. NON-RULE PROTOTYPES ON ACCURACY

To study the performance improvement of prototype learning due to impacts of rules, we compare the empirical effect of non-rule prototypes and rule prototypes. As shown in Figure 7, we found that more rule-prototypes can yield better accuracy in general, which shows learned rule-prototypes are better than non-rule prototypes.

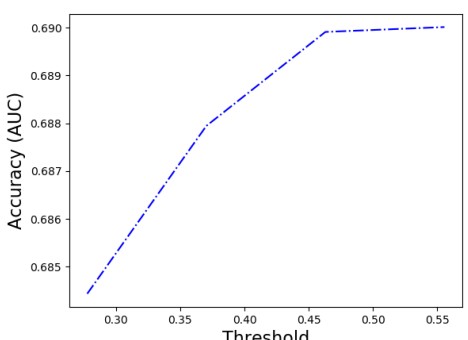

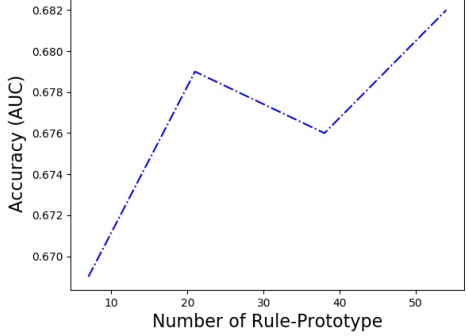

Figure 6: Effect of different threshold $\eta$ in Algorithm 1.

Figure 7: Effect of Fractions of Rule prototype suggest for the iterative learning, more rules can help prototype learning achieve better results.

### A.5 Prototype Rareness v.s. Accuracy

We then study the empirical effect of data reweighing threshold $\eta$ in Algorithm 1, where $\eta$ controls the number of prototypes from which the corresponding subjects are up-weighted. From Figure 6, we find that higher threshold usually corresponds to better accuracy.

### A.6 Performance on Benchmark Dataset

We added additional evaluation using "cars" and "breast" data from UCI data repository. For "cars" dataset, falling rule reach 80% classification accuracy, while PEARL reaches only 93% classification accuracy, both using 13 rules. Pure NN method achieve 95% classification accuracy. For "breast" dataset, falling rule method reach 85% classification accuracy while PEARL reaches 92%, both using 17 rules. Pure NN also reach 92%.

### A.7 Evaluation of learned Prototype

The maximal number of rules from rule learning is not set in advance; it is determined by data. In most cases, it produce at most 60 rules. We conduct experiment to study the average distance between data sample and prototype at the convergence state. The average distance is 0.643 for the method in Li et al. (2017) while for PEARL, the distance is 0.432, indicating that PEARL shows better clustering property than random prototype.

