# OpenReview forum: "Pearl: Prototype lEArning via Rule Lists"
_ICLR.cc/2019/Conference_

### Official Review · AnonReviewer3 · 2018-11-02
**Needs more thorough evaluation, more justification of design choices, and improvement in presentation clarity**

**Rating:** 4
**Confidence:** 4

**Review:**



Review Summary
--------------
The paper presents a combination of rule lists, prototypes, and deep representation learning to fit classifiers that are said to be simultaneously "accurate" and "interpretable". While the topic is interesting and the direction seems novel, I don't think the work is quite polished or competitive enough to be accepted without significant revision. The major issues include non-competitive evaluation of what "interpretability" means, ROC AUC numbers that are indistinguishable from standard deep learning (RCNN) pipelines that use many fewer parameters, and many unjustified choices inside the method itself. The paper itself could also benefit from revision to improve flow and introduce technical ideas to be more accessible to readers.


Paper Summary
-------------
The paper presents a new method called "PEARL" (Prototype Learning via Rule Lists), which produces a rule list, a set of prototypes, and a deep feed-forward neural network that can embed any input data into a low-dimensional feature space. The primary intended application is classifying subjects into a finite set of possible disorders given longitudinal electronic health records with categorical features observed at T irregular time intervals.

The paper suggests learning a representation for each subject's data by feeding the EHR time series into a recurrent convolutional NN. The input data is a 2 x T array, with one row representing observed data and second row giving time delay between successive observations. The vector output of an initial convolutional RNN is then fed into a highway network to produce a final vector denoted "h".

Given an encoder to produce feature vectors, and a fixed rule list learned from data itself, the paper suggests obtaining a prototype for each rule by computing the average vector of all data that matches the given rule. The quality of these prototypes and related neural networks (for computing features and predicting labels from features) is then assessed via their loss function in Eq. 1: a weighted combination of how well the prototypes match the learned embeddings (distance to closest prototype) and how well the classifier predicts labels.  The core idea is that the embedding is learned to classify well while creating a latent space that looks like the prototypes of the rule list.

After training an embedding and NN classifier on a fixed rule list, it seems the data is reweighted according to some heuristic procedure to obtain better properties, then a new rule list is trained and the process repeats again. (I admit the reweight procedure's purpose was never clear to me).

Experiments are done on a proprietary heart failure EHR dataset and on a subset of MIMIC data.

Strengths
---------
* Seems original: I'm unaware of any other method connecting rule lists AND prototypes AND NNs
* Neat applications to healthcare

Limitations
-----------
* Interpretability evaluation seems weak: no human subject experiments, no quantiative metrics, unclear if rule-lists shown is an apples-to-apples comparison
* Prototypes themselves never evaluated
* Many design choices inside method not justified with experiments -- why highway networks + RCNNs?

Major Issues with Method
------------------------

## M1: Not clear that AUC difference between PEARL and baselines is significant

The major issue is that the presented approach does not seem significantly different in predictive performance than the baseline Recurrent CNN. Comparing ROC AUC, we have PEARL's 0.688 to RCNN'S 0.682 with stddev of 0.009 on the proprietary heart failure dataset, and PEARL's 0.769 to RCNN's 0.766 with stddev of 0.009. When AUCs match this closely, I struggle to believe one model is definitively better, especially given that the RCNN has 2x *fewer* parameters (8.4k to 18.4k).

If the counterargument is that the resulting "deep model" is not "interpretable", one should at least compare to a post-processing step where the decision boundary of the RCNN is the reference to which a rule list or decision tree is trained.

## M2: Interpretability evaluation not clear.

Isn't the maximum number of rules set in advance?

Additionally, prototypes are a key part of this work, but the learned prototypes are not evaluated at all in any figure (except to track avg. distance from prototype while training). If prototypes are so central to this work, I would like to see a formal evaluation of whether the learned prototypes are indeed better (in terms of distance, or inspection of values by an expert, or something else) than alternatives like Li et al.

## M3: Missing a good synthetic/small dataset experiment

Neither of the presented data tasks is particularly easy to understand for non-experts. I'd suggest creating an additional experiment where the audience of ML readers is likely to easily grasp whether a set of rule lists is "good" for the problem at hand... maybe create your own synthetic task or a UCI dataset or something, or even use the stop-and-frisk crime dataset from the Angelino et al. 2018 paper. Then you can compare against just a few relevant baselines (rule lists only or prototypes only). I think a better illustrative experiment will help readers grasp differences between methods.

## M4: How crucial is feature selection?

In each iteration, Algo. 1 performs feature selection before learning rules. Are any other baselines (trees, rule lists) allowed feature selection before the classifier is learned? What would happen to PEARL without feature selection? What method is used for selection? (A search of the document only has 'feature selection' occur once, in the Alg. itself, so it seems explanation is missing).

## M5: Why are multiple algorithm iterations needed?

Won't steps 3 and 4 of Alg. 1 result in the same rules every time? It's not clear then why on subsequent iterations the algorithm would improve. Perhaps it's just the reweighting of data that causes these steps to change?

Minor issues
------------

## Loss function notation confusing

Doesn't the rule list classifier s_R take the data itself X? Not the learned embedding h(X)? Please fix or clarify Eq. 1. I think you might clarify notation by just writing yhat(h(X)) if you mean the predicted label of some example as done by your NNs. Using "R" makes folks think the rule list is involved.

## Not clear why per-example reweighting is required

None of the experiments assess why per-example reweighting (lines 6-9 of Algo. 1) is required. Readers would like to see a comparison of performance with and without this step.

## Not clear or justified when "averaged" prototypes are acceptable

Are your "averaged" prototypes guaranteed to satisfy the rule they represent? Is taking the average of vectors that match a rule always guaranteed to also match the rule? I don't think this is necessarily true. Consider a rule that says "if x[0] == 0 or x[1] == 0, then ___".  Suppose the only matching vectors are x_A = [0 1] and x_B = [1 0]. The average vector is [0.5 0.5] which doesn't work.

## Several different measures of distance used without careful justification

Why use two different distances -- Euclidean distance to assess distance to prototypes for prototype assignment, and then cosine similarity when deciding which examples to upweight or downweight? Why not just use Euclidean distance for both (appropriately transformed to a similarity)?

Comments on Presentation
------------------------
Overall I think every section of the paper needs significant revision to improve a reader's ability to understand main ideas. Notation could be introduced slowly (explain purpose and dimension of every variable), assumptions could be clearly stated (e.g. each individual rule can have ANDs but not ORs), and design choices justified. You might try the test of giving the paper to a colleague and having them explain back the ideas of each section to you... currently I do not believe this version passes this test.

The introduction claims that "clinicians are often unwilling to accept algorithm recommendations without clarity as to the underlying reasoning", but I would be careful in blindly asserting this without evidence. For a nice argument about avoiding blind assumptions about what doctor's will and won't accept, see Lipton's 2017 paper "The Doctor Just Won't Accept That" (https://arxiv.org/abs/1711.08037)

Additionally, the authors should clarify more precisely what definition of interpretability is needed for their applications. Is it simplicity? Is it conceptual alignment with known medical facts? Is it the ability to transparently list the rules in plain English?

Line-by-line details
--------------------

## Sec. 2

When introducing p_j, should clarify this this is one prototype vector of many.

When defining p_j = f_j(X), can you clarify what dimensionality p_j has? Is it always the same size as each example's data vector x_i?


## Sec. 3

Fig. 2: I don't find this figure very easy-to-understand. It's clear that after embedding raw features to a new space, the learned rules are *different*, but it's not clear they are *better*.  None of the illustrated rules perfectly segments the different colors, for example. I guess the point is all the red dots are within one rule? But they aren't alone (there are blue and orange dots too), so it's still not clear this would be a better classifier.

For EHR datasets, are you assuming that events are always categorical? And that outcomes "y" are always discrete (one-of-L) variables? Or could y be real-valued?

Eq. 1: You should make notation clearly indicate which terms depend on \theta. Currently it seems that nothing is a function of \theta.

Eq. 1: Do you also find the prototype set P that minimizes this objective? Or is there another way to obtain P given parameters \theta? This is confusing just from reading the eqn.

What size is the learned representation h(X)? Is it a vector?

Eq. 6: Do you really need a "network" to compute the distance to each of the K prototypes? Can't you just compute these distances directly?

## Sec 4

"Mac OS 1.4" : Do you mean Mac OS version 10.4? Not clear this is relevant.

4.3 Case Study: How do I read these rules? Is this rule applied only if ALL conditions are true? or if any individual one is true ("or")? This is unclear.

---

> ### Author Response · Authors · 2018-11-27
> **Improvement on evaluation, design choices and presentation**
>
> 1. AUC significance
> Our response: We agree that the AUC difference may not be significant. We have revised our paper and instead claim our method has comparable performance with neural network but has the interpretation power of rule lists.
>
>
> 2. evaluation of learned prototype
> Our response: No, the maximal number of rules from rule learning is not set in advance; it is determined by data. In most cases, it produce at most 60 rules. We conduct experiment to study the average distance between data sample and prototype at the convergence state. The average distance is 0.643 for the method in (Li et al 2018). While for PEARL, the distance is 0.432, indicating that PEARL shows better clustering property than random prototype. We have added these results to the appendix.
>
> 3. evaluation on benchmark data
> Our response: We thank the reviewer for this suggestion. We added additional evaluation using “cars” and “breast” data from UCI data repository. For “cars” dataset, falling rule reach 80% classification accuracy, while PEARL reaches only 93% classification accuracy, both using 13 rules.  Pure NN method achieve 95% classification accuracy. For “breast” dataset, falling rule method reach 85% classification accuracy while PEARL reaches 92%, both using 17 rules. Pure NN also reach 92%.  We have added these results to the appendix.
>
> 4. feature selection
> Our response: Falling rule method usually scales poorly in feature dimension. So we have to reduce the dimension and pick some “important” feature to find the rule. In this paper, we use chi-2 test, a standard feature selection method to select best features. Empirical studies show that the final performance is not sensitive to different feature selection methods as available at sklearn (compared methods are variance-based feature, univariate feature with chi-2 test, recursive feature elimination, and L-1 based feature selection). We have addressed this issue in Section 3.1.1.
>
> 5. multi iteration and per-example weights
> Our response: No, the rules are different because the input data to the rule learning module are different at each iteration. Multiple iteration does improve the performance compared single iterate, as shown in last two rows in Table 3.
>
> 6. Loss function
> Our response:  s_R is not the rule list classifier, but is a neural network architecture from the prediction module which uses rule list information.
>
> 7. “Average” Prototype learning
> Our response: We agree that the learned prototypes by averaging data samples not necessarily satisfy rules. However, it is not a problem in PEARL as we do not need hard satisfiability. The average of positive data samples within each prototype is simply used to regularize the learning (seen in Section 3.1.2), helping to learn a better representation in NN.
>
>
> 8. consistent distance metric:
> Our response: We thank the reviewer for the suggestion. We have revised our experiment to uses cosine similarity in prototype layer instead of Euclidean distance, as shown in Eq 6. The experimental results are correspondingly updated. The results show no big difference compared with previous results.
>
>
> 9. We have addressed paper presentation, fixed notation,  added relevant citations,  and clarified line-by-line comments in the revision, and here are some response to the major ones:
> 1) p_j and h(x) are vectors of the same predefined dimension, which is a hyperparameter.
> 2) Equation 6: the distance metric is computed in the learned representation space, rather than the original space. The representation space that h(X) lies in should be more discriminative.
> 3) Eq. 1: prototype set P is determined given rule list R and learned representation h(X) hence there is no new parameter. We have modified Equation 1 to reflect so and clarified this point below Equation 1.
> 4) Case Study 4.3: this rule is applied only if all conditions are true. We have clarified it in the writing.
> 5) Fig. 2: we have redrawn the figure to show a better classifier per suggestion.

---

### Official Review · AnonReviewer1 · 2018-11-02
**Interpretability insufficiently defined**

**Rating:** 3
**Confidence:** 3

**Review:**

This paper aims at tackling the lack of interpretability of deep learning models, which is especially problematic in a healthcare setting --the focus of this research paper. Specifically, the authors propose Prototype lEArning via Rule Lists (PEARL), which combines rule learning and prototype learning to achieve more accurate classification and better predictive power than either method independently and which the authors claim makes the task of interpretability simpler.
The authors present an interesting and novel architecture in PEARL. Combining the two approaches of rule lists and prototype learning. However, my main concern with the paper and with the architecture in general is the lack of clarity upfront regarding what the authors perceive as the criteria for interpretability. This seems to be one of the chief aims of the paper, however, the authors don’t reach this point until Section 4 of the paper. Given that this is one of the main strengths of the paper as proposed by the authors, this needs to be given more prominence and also needs to be made more explicit what the authors mean by this. The authors define interpretability as measured by the number of rules and number of protoypes identified by a particular model, without, providing an argument, justification, or a citation of previous work which justifies these criterion. Especially since this is one of the main points of the paper, this needs to be better argued and the authors should either elaborate on this point, or restrain on making claims that these models are more interpretable.
The model architecture of Section 3.1 was quite obscure both from the intuitive and implementation level. It’s not clear how the different modules (prototype learning, rule lists) link together in practice, nor how these come together to create an interpretable model.
Generally, the paper is quite poorly structured and there were several grammatical errors which made the paper quite hard to follow. Although the problems articulated are important, the paper did not do sufficient justice to addressing these problems.

---

> ### Author Response · Authors · 2018-11-27
> **adding definition of interpretability**
>
> 1. the definition of interpretability
> Our response: In the revised draft, we add a definition of interpretability per suggestion in Section 2. Following (Lakkaraju et al 16), we mainly tackle the size (by combining rule in rule lists into prototypes) and the length (by replacing the clauses in each rule using a prototype).
>
> 2. model choices of deep networks
> Our response: PEARL has two modules combining conventional prediction with NN and rule-list classifiers. The prediction components consist of a 1-dimensional CNN (convolution over temporal dimension), then an RNN (temporal modeling), followed by a highway network (alleviate gradient vanishment issue) These components are standard in classification and prediction tasks. By combing the power of RNN and CNN, we can utilize both temporal information and the extracted abstract features for prediction. After RNN and CNN, a prototype layer is used to combine NN and rule-list learning via data re-weighting. The main intuition is that we would like NN to learn and produce prototypes that are closely related to rules in the rule lists, with one-to-one or many-to-one rule-prototype mapping. This serves as a constraint to make each prototype become a replacement of clauses in each rule, transforming “if x satisfies p, then  x = q” to “if x is close to prototype, then x = q”.
>
> 3. paper organization
> Our response: We have revised the paper, with major updates in Section 2 and 3 to introduce a complete definition of interpretability and an overview of PEARL along some intuition of its components choices. We thank the reviewers for the suggestion.

---

### Official Review · AnonReviewer2 · 2018-11-11
**Interesting idea but needs significant improvement in terms of presentation and design of method and experiments**

**Rating:** 5
**Confidence:** 4

**Review:**

Summary:
This paper presents a new interpretable prediction framework which combines rule based learning, prototype learning, and NNs. The method is particularly applicable to longitudinal data. While the idea of bringing together rules, prototypes, and NNs is definitely novel, the method itself has some unclear design choices. Furthermore, the experiments seem pretty rudimentary and the presentation can be significantly improved.

Detailed Comments:
1. In Section 2, the authors seem to define rule list as a set of independent if-then rules. Please note that rule lists have an "else if" clause which creates a dependency between the rules. Please refer to "Interpretable decision sets" by Lakkaraju et. al. for understanding the differences between rule lists and rule sets.
2. Section 3.1 is quite confusing. It would be good to give an intuition as to how the various pieces are being combined and in why it makes sense to combine them in this way. The data reweighting process seems a bit adhoc to me. What other choices for reweighting were considered?
3. I would strongly encourage the authors to carry out at least a simple user study before claiming that the proposed method is more interpretable than existing rule lists. Adding both prototypes and rules, in fact, adds to the cognitive burden of an end user - it would be interesting to see when and how having both prototypes and rules will help an end user.

Pros:
1. First approach to combine NNs, rule learning, prototype learning
2. Provides an interpretable method for predictions on longitudinal medical data
3. Experimental results seem to suggest that the proposed approach is resulting in accurate and interpretable models.

Cons:
1. The various pieces in the method (rule learning, prototype, NNs, data reweighting) seem to be somewhat haphazardly connected. Section 3.1 does not give me a good idea about how the different pieces are resulting in an accurate and interpretable model
2. The paper makes claims such as "Experimental results also show the resulting interpretation
of PEARL is simpler than the standard rule learning." without actually doing any significant user studies. Furthermore, any other synthetic data experiments which could demonstrate the various facets of accuracy-interpretability tradeoffs are missing
3. The presentation of the paper is quite unclear. See detailed comments above.

---

> ### Author Response · Authors · 2018-11-27
> **big modification on presentation, method and experiment**
>
> 1.  rule list definition
> Our response: The definition does not imply independent rules, as seen by previous work (Figure 2, Angelino et al 2018). We have added a  sentence to clarify this point in Section 2 and mention related definition such as (Lakkaraju et al 16).
>
> 2. overall system design
> Our response: PEARL has two modules combining conventional prediction with NN and rule-list classifiers. The prediction components consist of an 1-dimensional CNN (convolution over temporal dimension), then an RNN (temporal modeling), followed by a highway network (to alleviate gradient vanishment issue). These components are standard in classification and prediction tasks. After RNN and CNN, a prototype layer is used to combine NN and rule-list learning via data re-weighting. The main intuition is that we would like NN to learn and produce prototypes that are closely related to rules in the rule lists, with one-to-one or many-to-one rule-prototype mapping. This serves as a constraint to make each prototype as a surrogate for clauses in each rule, transforming “if x satisfies z, then  x = q” to “if x is close to a prototype p, then x = q”. Moreover, the data reweighting is the key to combine rule list and NN classifiers, and we do not know any alternative.
>
> 3. user study
> Our response: we believe the user study task is not really helpful here, since the goal of PEARL is to learn more accurate and smaller rule lists (compared with traditional rule lists), which has been shown in the experiments.  It does not need a study to conclude that it would ease the cognitive burden of human. In essence, PEARL produces a rule list with prototype as clauses. They can be presented to a typical rule list as seen by Section 4.3, which does not increase cognitive burden in any way than existing rule lists. Moreover, as illustrated in Fig 3, PEARL can maintain good AUC performance even with a much smaller number of rule (< 5), but conventional rule learning (Angelino 2018) performs poorly even when all 50 rules are used.
>
> 4. Synthetic datasets:
> Our response: we have added a few synthetic dataset results in the appendix for comparison. Overall, our method has comparable performance as deep models, but more interpretable, and better performance than conventional rule lists.

---

### Author Response · Authors · 2018-11-27
**Summary of major improvements**

We thank the reviewers for the constructive comments and suggestions. We have substantially revised the paper, in particularly
1) adding a precise definition of interpretability,
2) clarifying system design,
3) improving the presentation of the paper,
to respond to all the question raised. Hopefully the revised paper has clarified some confusion and addressed all the concerns from reviewers. Our detailed responses to the comments from each reviewer are listed below.

---

### Meta-Review · Area_Chair1 · 2018-12-16
**Presentation and Evaluation concerns remain**

**Confidence:** 4
**Recommendation:** Reject

**Metareview:**

This paper presents an approach that combines rule lists with prototype-based neural models to learn accurate models that are also interpretable (both due to rules and the prototypes). This combination is quite novel, the reviewers and the AC are unaware of prior work that has combined them, and find it potentially impactful. The experiments on the healthcare application were appreciated, and it is clear that the proposed approach produces accurate models, with much fewer rules than existing rule learning approaches.

The reviewers and AC note the following potential weaknesses: (1) there are substantial presentation issues, including the details of the approach, (2) unclear what the differences are from existing approaches, in particular, the benefits, and (3) The evaluation lacked in several important aspects, including user study on interpretability, and choice of benchmarks.

The authors provided a revision to their paper that addresses some of the presentation issues in notation, and incorporates some of the evaluation considerations as appendices into the paper. However, the reviewer scores are unchanged since most of the presentation and evaluation concerns remain, requiring significant modifications to be addressed.